# Chlorhexidine and octenidine susceptibility of bacterial isolates from clinical samples in a three-armed cluster randomised decolonisation trial

**Luisa A. Denkel**[1,2]*, **Tobias S. Kramer**[1,2], **Frank Schwab**[1,2], **Jennifer Golembus**[1,2], **Solvy Wolke**[1,2], **Petra Gastmeier**[1,2], **Christine Geffers**[1,2]

**1** Institute of Hygiene and Environmental Medicine, Charité Universitätsmedizin Berlin, Corporate Member of Freie Universität Berlin, Humboldt-Universität zu Berlin and Berlin Institute of Health, Berlin, Germany, **2** National Reference Center for the Surveillance of Nosocomial Infections, Charité Universitätsmedizin Berlin, Corporate Member of Freie Universität Berlin, Humboldt-Universität zu Berlin and Berlin Institute of Health, Berlin, Germany

* luisa.denkel@charite.de

**Data Availability Statement:** All relevant data are within the manuscript and its Supporting Information files.

## Abstract

### Background

Routine use of chlorhexidine or octenidine for antiseptic bathing may have unintended consequences. Our analysis aimed to assess the phenotypic susceptibility of bacterial isolates from clinical samples to chlorhexidine and octenidine collected from intensive care units (ICU) that routinely used 2% chlorhexidine-impregnated wash cloths or 0.08% octenidine wash mitts (intervention) or water and soap (control) for daily patient care.

### Methods

This study was conducted within the context of a three armed cluster-randomised controlled decolonisation trial (Registration number DRKS00010475, registration date August 18, 2016). Bacterial isolates were collected prior to and at the end of a 12-month-intervention period from patients with ≥ 3 days length of stay at an ICU assigned to one of two intervention groups or the control group. Phenotypic susceptibility to chlorhexidine and octenidine was assessed by an accredited contract research laboratory determining minimal inhibitory concentrations (MIC) as percentage of extraction solutions used. MIC were reported as estimated concentrations in µg/ml derived from the chlorhexidine and octenidine extraction solutions. Statistical analyses including generalized estimating equation models were applied.

### Results

In total, 790 ICU-attributable bacterial isolates from clinical samples (e.g. blood, urine, tracheal aspirate) were eligible for all analyses. Pathogens included were *Staphylococcus aureus* (n = 155), coagulase-negative staphylococci (CoNS, n = 122), *Escherichia coli* (n = 227), *Klebsiella* spp. (n = 150) and *Pseudomonas aeruginosa* (n = 136). For all species,

**Funding:** This study was funded by the German Federal Ministry of Education and Research within the scope of the InfectControl consortium (Grant No. 03ZZ0807A) awarded to PG. Sage Products / Stryker and Schülke, funded the (antiseptic) products for the intervention and supported the investigation of tolerances to chlorhexidine and octenidine by an independent accredited contract laboratory. Dr. Brill + Partner GmbH was the independent contract-laboratory assigned by Schülke and Stryker /Sage to conduct the investigations of tolerances to chlorhexidine and octenidine (by minimum inhibitory concentration testing). The funders had no role in study design, data collection and analysis, decision to publish, or preparation of the manuscript.

**Competing interests:** The authors have read the journal's policy and have the following competing interests: Sage Products / Stryker and Schülke and provided support in the form of funding for this study. Dr. Brill + Partner GmbH was the independent contract-laboratory assigned by Stryker /Sage Products and Schülke to conduct the investigations of tolerances to chlorhexidine and octenidine (by minimum inhibitory concentration testing). There are no patents, products in development or marketed products associated with this research to declare. This does not alter our adherence to PLOS ONE policies on sharing data and materials.

chlorhexidine and octenidine MIC did not increase from baseline to intervention period in the antiseptic bathing groups. For proportions of bacterial isolates with elevated chlorhexidine / octenidine MIC ($\geq$ species-specific chlorhexidine / octenidine $MIC_{50}$), adjusted incidence rate ratios (aIRR) showed no differences between the intervention groups and the control group (intervention period).

## Conclusion

We found no evidence for reduced phenotypic susceptibilities of bacterial isolates from clinical samples to chlorhexidine or octenidine in ICUs 12 months after implementation of routine antiseptic bathing with the respective substances.

## Introduction

Chlorhexidine gluconate and octenidine dihydrochloride are two cationic biocides that can bind covalently to the bacterial cell membrane, subsequently cause depolarization and can lead to bacterial cell death if used at high concentrations [1, 2]. Meta-analyses and clinical trials are available that demonstrate the effectiveness of chlorhexidine bathing to reduce healthcare-associated infections (HAI) including bloodstream infections (BSI) in intensive care units (ICUs) [3–9]. Thus, chlorhexidine is most commonly used for antiseptic bathing of intensive care patients. At the same time, only few clinical trials on the effect of antiseptic bathing with octenidine for BSI prevention are available [10–12]. In Germany, antiseptic bathing of ICU patients is no routine infection control practice. If applied, however, octenidine is the most frequently used substance [13].

We conducted a three-armed cluster-randomised controlled trial (cRCT) in 72 adult ICUs to investigate the effect of daily patient bathing with chlorhexidine, octenidine or routine care with water and soap (control) on central-line associated BSI (CLABSI) rates [14]. This trial found no preventive effect of antiseptic bathing neither with chlorhexidine nor octenidine when comparing CLABSI rates of the intervention groups with the control group [14]. The post hoc before-after analysis of our cluster-randomised decolonisation trial, however, suggests that ICUs with CLABSI rates $\geq$0.8 CLABSI per 1000 central-line (CL) days might benefit from bathing with 2% chlorhexidine-impregnated cloths [15]. At the same time, octenidine showed no preventive effect on ICU-attributable CLABSI [15].

The use of chlorhexidine or octenidine may have unintended consequences. High usage of these antiseptic substances is associated with increases in chlorhexidine and octenidine minimal inhibitory concentrations (MIC) among clinical isolates [16]. Further, long-term use of chlorhexidine is suspected not only to enhance the development of non-susceptibility, but also to increase antibiotic cross-resistance, decolonization failure or potentially disadvantageous alterations of the skin microbiome [17–20].

Our analysis aimed to answer the question whether daily patient bathing with 2% impregnated chlorhexidine cloths or 0.08% octenidine wash mitts might lead to reduced chlorhexidine or octenidine susceptibilities of ICU-attributable bacterial isolates from clinical samples in the CLIP-ID trial.

## Methods

### Trial design and participants

This investigation was done within a cluster-randomised controlled trial (cRCT) as part of the project Climate and pathogens–impact of decolonisation (CLIP-ID, registration number DRKS00010475, registration date August 18, 2016). Details of the CLIP-ID trial including the study protocol were published elsewhere [14]. Briefly, our cRCT was conducted on 72 ICUs in 68 hospitals in Germany and Austria. Each ICU represented one cluster and was randomly assigned to one of three decolonisation regimes that had to be applied for 12 consecutive months. The random allocation sequence was computer-generated and applied on the cluster (ICU) level. We stratified ICUs by type of ICU (medical, surgical, interdisciplinary wards in hospitals with < 400 beds and interdisciplinary wards in hospitals with ≥ 400 beds) and size of hospital before randomization. In the chlorhexidine group, all ICUs performed daily patient bathing with 2% chlorhexidine-impregnated cloths (Sage 2% Chlorhexidine Gluconate (CHG) by Stryker) below the jaw line and chlorhexidine-free cloths above the jaw line (Stryker). In the octenidine group, ICUs used 0.08% octenidine disposable wash mitts (Octenisan® by Schülke). In the routine care (control) group, all ICUs applied non-antiseptic soap and water (routine care) for daily patient care.

### Collection of the ICU-attributable bacterial isolates from clinical samples

All 72 ICUs participating were requested to collect 10 bacterial isolates from clinical samples prior to and 10 bacterial isolates at the end of the intervention period to analyze phenotypic susceptibility to the antiseptic substances (chlorhexidine and octenidine) applied in this trial. Bacterial isolates collected must met the following inclusion criteria: i) specimen were collected for medical or diagnostic purposes only; ii) bacteria isolated from clinical material (e.g. tracheal aspirate, urine, blood, liquor) not colonization sites (no screening isolates from rectal swabs, stool, nasal-pharyngeal swabs); iii) clinical material from patients with length of stay of at least 3 days on the ICU participating and iv) identification of *Staphylococcus (S.) aureus*, *Enterobacterales* (e.g. *Escherichia (E.) coli*, *Klebsiella (K.)* subspecies (spp.) or *Pseudomonas (P.) aeruginosa* from all clinical material or coagulase-negative staphylococci (CoNS) from blood samples only. Bacterial isolates (with Amies transport medium) were sent by the diagnostic laboratories of ICUs participating in the trial to the coordinating study center. The study was not blinded. However, collection of bacterial isolates was done by laboratory personnel who were not informed about assignments of ICUs to one of three study groups. Information on isolates (e.g. ICU name, ICU day of sample collection, anonymous sequential patient number, date, material, and species) were provided by the ICU and their diagnostic laboratories in anonymous form. No patient based data were collected or distributed.

**Phenotypic susceptibility to chlorhexidine and octenidine.** Phenotypic susceptibility of bacterial isolates from clinical samples to the antiseptic substances contained in the chlorhexidine-impregnated cloths and octenidine disposable wash mitts were analysed by determining minimal inhibitory concentrations (MIC). Phenotypic susceptibility testings were assigned as contract work to an accredited laboratory and conducted according to VAH method 7 (A00040) [21].

Extraction solutions of antiseptic substances (chlorhexidine and octenidine) were harvested by wringing out chlorhexidine-impregnated cloths (Stryker / Sage Products) and octenidine wash mitts (Schülke) provided by the manufacturer. Concentrations of these extraction solutions were estimated to be similar to the concentrations of the ready-to-use products reported by the manufacturers: 2% (20,000 μg/ml) for chlorhexidine and 0.08% (800 μg/ml) for

octenidine. Dilutions of antiseptic substances were made in water of standardized water hardness (WSH) according to VAH method 5 [21]. Briefly, bacteria were inoculated in soybean casein digest agar (CSA) at 37°C for 18 – 24h. Concentrations of these bacterial solutions were adjusted to 1.5–5.0 x $10^8$ colony forming units (CFU)/ml. Subsequently, 150 μl of the respective antiseptic dilutions and 150 μl doubled concentrated soybean casein solutions (CSL) were mixed. Microtiter plates were inoculated with 3 μl bacteria (1:10-dilution in CSL) yielding in a final bacterial concentration of 1.5–5.0 x $10^5$ colony forming units (CFU)/ml. Bacteria were inoculated at 37°C for 48 h. The lowest concentration of extraction solutions without bacterial growth (detectable turbidity) was interpreted as minimal inhibitory concentration (MIC) reported in percentage of extraction solution. Percentages were converted to estimated (est.) concentrations of extraction solutions in μg / ml. Controls were inoculated with WSH instead of antiseptic dilutions. Determination of bacteriostatic efficacy and suitable neutralizing agents was done according to VAH method 7 [21]. The following reference strains were included to the analysis: *Staphylococcus aureus* ATCC6538 (DSM 799), *Proteus mirabilis* ATCC 14153 (DSM 788), *Klebsiella pneumoniae* ESBL (DSM 16609), *Pseudomonas aeruginosa* ATCC 15442 and *Escherichia coli* NCTC 10538.

**Statistical analyses.** In the descriptive analysis, numbers with percentages, medians ($MIC_{50}$) with interquartile ranges (IQR) and / or percentiles ($MIC_{90}$) were calculated. The variables "sample site" and "study group and study period" were dummy-coded. P-values were calculated by Chi-square test or Fisher's exact test for categorical, and by Mann-Whitney U Test for continuous variables.

We generated a binary variable to represent phenotypic chlorhexidine and octenidine susceptibilities of bacterial isolates from clinical samples. All bacterial isolates were categorized according to their chlorhexidine and octenidine MIC being $\geq$ or $<$ the species-specific chlorhexidine and octenidine $MIC_{50}$ (identified in this study). Bacterial isolates were categorized as "yes" (bacterial isolates with chlorhexidine / octenidine susceptibility $\geq$ species-specific $MIC_{50}$) or "no" (chlorhexidine /octenidine susceptibility of bacterial isolates $<$ species-specific $MIC_{50}$).

In the multivariable analysis, we applied generalised estimating equation (GEE) models to identify effects of chlorhexidine / octenidine interventions on the susceptibility of bacterial isolates from clinical samples to these substances. We used one GEE model for all species to investigate the susceptibility to chlorhexidine, and another GEE model to investigate the susceptibility to octenidine. The outcomes "phenotypic susceptibilities to chlorhexidine / octenidine" were represented by proportions of bacterial isolates with chlorhexidine or octenidine MIC being $\geq$ the species-specific chlorhexidine / octenidine $MIC_{50}$. Interaction of study groups (chlorhexidine / octenidine / control) and study periods (baseline / intervention) as well as potential cluster effects were considered in these models [22]. Control group (intervention period) served as reference. Further, ICU day of sample collection and clinical site (blood, tracheal aspirate, urine, wound, or other) were determined as possible confounders and considered in all models. The category "other" as clinical site included e.g. intra-operative samples, intra-abdominal swabs and / or tissue. All parameters added one degree of freedom to the model.

P-values less than 0.05 were considered significant. All analyses were performed using SPSS 27 (IBM SPSS statistics, Somer, NY, USA).

**Sample size calculation.** Initial sample size calculation of this cRCT was done for the primary outcome CLABSI [14]. However, we performed a power calculation for the outcomes "phenotypic susceptibilities to chlorhexidine / octenidine" represented by proportions of bacterial isolates with chlorhexidine or octenidine MIC being $\geq$ the species-specific chlorhexidine / octenidine $MIC_{50}$. The latter were identified in this study (Table 2). We assumed that at the

end of the intervention period, 35% of isolates were $\geq \text{MIC}_{50}$ in the control group (routine care) compared to 65% in the intervention group (antiseptic bathing with chlorhexidine or octenidine). Further, we assumed a ratio of 1:1 between the groups. With individual randomization, a sample size of 70 bacterial isolates in each group would have been required to show the difference with a power of 80%, and a two-sided type 1 error of 0.025. Adjusting for cluster effects with 13 clusters per arm and an intracluster correlation coefficient of 0.1 (adapted from literature [23]), we must have included 101 bacterial isolates per arm. Multiple comparisons to the routine care group were considered by a two-sided type 1 error of 0.025.

## Ethics approval

The institutional ethical review board (IRB) of the Charité Universitätsmedizin Berlin granted their approval including a waiver of informed consent for this trial (processing number EA1/ 093/16). The present study did not include any individual patient data, but analysed aggregated and anonymous data only.

## Results

Intervention periods lasted from February 1, 2017 to January 31, 2018 in the octenidine and control group, and from June 1, 2017 to May 31, 2018 in the chlorhexidine group. Baseline periods included 12 months before the interventions started in each study group.

In total, 996 bacterial isolates from clinical samples were collected from participating wards. Among them, 797 bacterial isolates were eligible for analyses of susceptibility to chlorhexidine and octenidine (Fig 1).

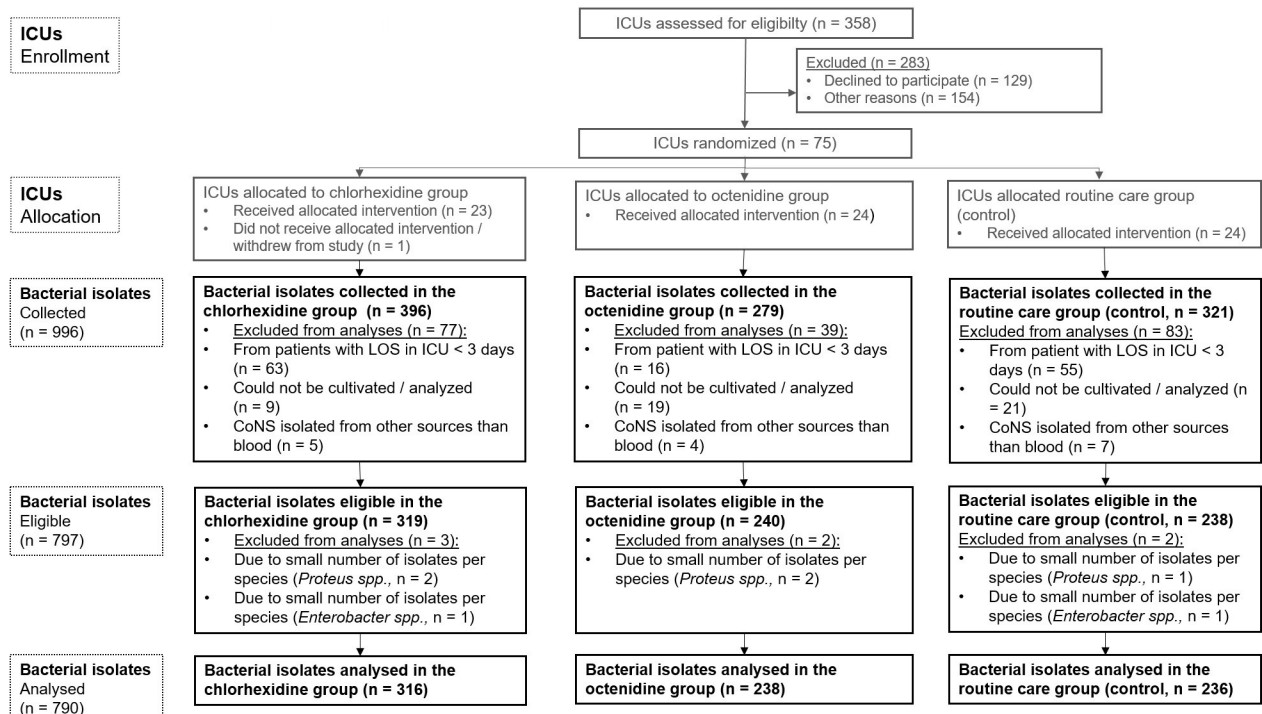

**Fig 1. Flow chart of ICU-attributable bacterial isolates from clinical samples eligible for analyses of susceptibilities to chlorhexidine and octenidine and included to descriptive analyses.** CoNS, coagulase-negative staphylococci. ICU, intensive care unit. spp., subspecies.

The selected pathogens detected from ICU-attributable clinical isolates were *E. coli* (n = 227), *S. aureus* (n = 155), *Klebsiella spp.* (n = 150), *P. aeruginosa* (n = 136), CoNS (n = 122), *Proteus spp.* (n = 5) and *Enterobacter spp.* (n = 2). Due to the small number of isolates per species, *Proteus spp.* and *Enterobacter spp.* were excluded from all further analyses (Fig 1). In consequence, 790 clinical isolates from 60 ICUs were included in the descriptive analyses. Among them, 417 isolates were collected during baseline when all wards used water and soap for daily bathing, and 373 isolates were collected at the end of the intervention period. Clinical isolates from baseline and intervention period were available from 42 of the 60 ICUs (70.0%) responding.

Characteristics of bacterial isolates reported by study group and period are shown in Table 1. No differences in sample site or ICU day of sample collection were detected between study groups neither during baseline nor intervention period (Table 1). The same was true for baseline versus intervention period comparisons in each study group except for the chlorhexidine group. In this group, the samples sites differed between baseline and intervention periods for *E. coli* and *Klebsiella spp.* (Table 1).

## Phenotypic susceptibility of bacterial isolates from clinical samples to chlorhexidine and octenidine

**$MIC_{50}$ and $MIC_{90}$.** For all species, phenotypic susceptibility to chlorhexidine depicted as $MIC_{50}$ and $MIC_{90}$ in est. µg/ml did not increase comparing baseline and the intervention periods in all study groups, as well as comparing intervention groups with control group at the end of the intervention (Table 2). The same was true for phenotypic susceptibility to octenidine (Table 2).

Phenotypic susceptibility to chlorhexidine and octenidine of commercially available reference strains (*S. aureus*, *E. coli*, *P. aeruginosa*, *K. pneumoniae* and *Proteus mirabilis*) are shown in S1 Table. Chlorhexidine and octenidine $MIC_{50}$ with IQR and $MIC_{90}$ in percentage of stock solution are given in S2 Table.

## Distribution of chlorhexidine and octenidine MIC for all ICU-attributable bacterial isolates from clinical samples

In the intervention groups, the percentages of bacterial isolates with higher chlorhexidine MIC indicating reduced chlorhexidine susceptibility did not increase comparing baseline and intervention periods for all species (Fig 2A–2E). Further, no increase was observed comparing intervention groups with control group (at the end of the intervention period) for all species. The same observations were made for octenidine MIC (Fig 2A–2E).

Interestingly, ICUs in the control group that continued bathing with water and soap collected some bacterial isolates with high chlorhexidine and / or octenidine MIC during intervention period even though no antiseptic substances were applied. High chlorhexidine MIC were found in the baseline period and / or in control group for some isolates of *S. aureus*, *E. coli*, *Klebsiella spp.* and *P. aeruginosa* and high octenidine MIC for *P. aeruginosa* (Fig 2A and 2C–2E).

Among the 155 *S. aureus* isolates, 23 (14.8%) had chlorhexidine MIC $\geq$ 8µg/ml, while this was the case for 10 of 122 CoNS (8.2%). High chlorhexidine MIC ($\geq$ 64 µg/ml) were rare in *E. coli* and *Klebsiella* spp. but frequent in *P. aeruginosa* ($\geq$ 50 µg/ml, Fig 2C–2E).

Characteristics of all 23 *S. aureus* isolates with chlorhexidine MIC $\geq$ 8µg/ml can be found in S3 Table.

**Table 1. Characteristics of ICU-attributable bacterial isolates from clinical samples included in the analyses (n = 790) in total and reported by study group and period.**

| | Total | Chlorhexidine group | | P-value[a] | Octenidine group | | P-value[a] | Routine care group = control | | P-value[a] |
|---|---|---|---|---|---|---|---|---|---|---|
| | | Baseline | Intervention | | Baseline | Intervention | | Baseline | Intervention | |
| **Staphylococcus aureus** | | | | | | | | | | |
| n (%) | 155 (100.0) | 38 (25.0) | 22 (14.2) | | 24 (15.5) | 18 (11.5) | | 23 (14.7) | 30 (19.2) | |
| ICU day of sample collection[b], median (IQR) | 6 (4–11) | 6 (4–8) | 6 (4–9) | 0.769 | 7 (5–20) | 6 (4–9) | 0.084 | 6 (4–11) | 5 (4–9) | 0.842 |
| **Sample site** | | | | | | | | | | |
| Blood[b], n (%) | 27 (17.3) | 7 (17.9) | 4 (18.2) | 0.633 | 3 (12.5) | 2 (11.1) | 0.639 | 3 (13.0) | 8 (26.7) | 0.193 |
| Tracheal aspirate[b], n (%) | 87 (55.8) | 23 (59.0) | 16 (72.2) | 0.340 | 14 (58.3) | 11 (61.1) | 0.856 | 10 (43.5) | 13 (43.3) | 0.992 |
| Urine[b], n (%) | 4 (2.6) | 1 (2.6) | 0 (0.0) | 0.633 | 0 (0.0) | 1 (5.6) | 0.420 | 0 (0.0) | 2 (6.7) | 0.316 |
| Wound[b], n (%) | 18 (11.6) | 4 (10.3) | 1 (4.5) | 0.389 | 3 (12.5) | 1 (5.6) | 0.420 | 5 (21.7) | 4 (13.3) | 0.328 |
| Others[b], n (%) | 19 (12.3) | 3 (7.7) | 1 (4.5) | 0.532 | 4 (16.7) | 3 (16.7) | 0.665 | 5 (26.3) | 3 (15.8) | 0.213 |
| **Coagulase-negative staphylococci** | | | | | | | | | | |
| n (%) | 122 (100.0) | 31 (25.4) | 31 (25.4) | | 17 (13.9) | 11 (9.0) | | 15 (12.2) | 17 (13.9) | |
| ICU day of sample collection[b], median (IQR) | 11 (6–19) | 13 (8–18) | 10 (5–21) | 0.418 | 17 (9–19) | 10 (3–16) | 0.154 | 11 (6–24) | 7 (4–17) | 0.363 |
| **Sample site** | | | | | | | | | | |
| Blood[b], n (%) | 122 (88.4) | 31 (100.0) | 31 (100.0) | n.d. | 17 (100.0) | 11 (100.0) | n.d | 15 (100.0) | 17 (100.0) | n.d. |
| Tracheal aspirate[b], n (%) | 0 (0.0) | 0 (0.0) | 0 (0.0) | n.d. | 0 (0.0) | 0 (0.0) | n.d. | 0 (0.0) | 0 (0.0) | n.d. |
| Urine[b], n (%) | 0 (0.0) | 0 (0.0) | 0 (0.0) | n.d. | 0 (0.0) | 0 (0.0) | n.d. | 0 (0.0) | 0 (0.0) | n.d. |
| Wound[b], n (%) | 0 (0.0) | 0 (0.0) | 0 (0.0) | n.d. | 0 (0.0) | 0 (0.0) | n.d. | 0 (0.0) | 0 (0.0) | n.d. |
| Others[b], n (%) | 0 (0.0) | 0 (0.0) | 0 (0.0) | n.d. | 0 (0.0) | 0 (0.0) | n.d. | 0 (0.0) | 0 (0.0) | n.d. |
| **Escherichia coli** | | | | | | | | | | |
| n (%) | 227 (100.0) | 64 (28.2) | 37 (16.3%) | | 41 (18.1) | 18 (7.9%) | | 39 (17.2) | 28 (11.9) | |
| ICU day of sample collection[b], median (IQR) | 11 (6–19) | 10 (5–16) | 7 (4–20) | 0.584 | 9 (5–20) | 11 (5–15) | 0.947 | 7 (4–14) | 12 (4–22) | 0.154 |
| **Sample site** | | | | | | | | | | |
| Blood[b], n (%) | 16 (7.1) | 2 (3.1) | 5 (13.5) | 0.096 | 3 (7.3) | 1 (5.6) | 1.000 | 2 (5.1) | 3 (10.7) | 0.642 |
| Tracheal aspirate[b], n (%) | 87 (38.3) | 30 (46.9) | 8 (21.6) | 0.012* | 16 (39.0) | 8 (44.4) | 0.457 | 13 (33.3) | 12 (42.9) | 0.427 |
| Urine[b], n (%) | 50 (22.0) | 8 (12.5) | 13 (35.1) | 0.007* | 11 (26.8) | 4 (22.2) | 1.000 | 10 (25.6) | 4 (14.3) | 0.259 |
| Wound[b], n (%) | 33 (14.5) | 8 (12.5) | 6 (16.2) | 0.405 | 3 (7.3) | 4 (22.2) | 0.184 | 8 (20.5) | 4 (14.3) | 0.512 |
| Others[b], n (%) | 41 (18.1) | 16 (25.0) | 5 (13.5) | 0.131 | 8 (19.5) | 1 (5.6) | 0.252 | 6 (15.4) | 5 (17.9) | 1.000 |
| **Klebsiella spp.** | | | | | | | | | | |
| n (%) | 150 (100.0) | 28 (18.7) | 28 (18.7) | | 23 (15.3) | 29 (19.3) | | 20 (13.3) | 22 (14.7) | |
| ICU day of sample collection[b], median (IQR) | 10 (5–20) | 11 (6–26) | 8 (5–24) | 0.565 | 12 (6–21) | 11 (6–19) | 0.712 | 9 (4–20) | 9 (5–16) | 0.889 |
| **Sample site** | | | | | | | | | | |
| Blood[b], n (%) | 18 (12.0) | 3 (10.7) | 5 (17.9) | 0.705 | 1 (4.3) | 2 (6.9) | 1.000 | 4 (20.0) | 3 (13.6) | 0.691 |
| Tracheal aspirate[b], n (%) | 74 (49.3) | 9 (67.0) | 15 (53.6) | 0.105 | 12 (52.2) | 17 (58.6) | 0.642 | 9 (45.0) | 12 (54.5) | 0.537 |
| Urine[b], n (%) | 17 (11.3) | 5 (17.9) | 5 (17.9) | 1.000 | 2 (8.7) | 3 (10.3) | 1.000 | 2 (10.0) | 0 (0.0) | 0.221 |
| Wound[b], n (%) | 21 (11.3) | 1 (3.6) | 2 (7.1) | 1.000 | 5 (21.7) | 6 (20.7) | 1.000 | 3 (15.0) | 4 (18.2) | 1.000 |
| Others[b], n (%) | 20 (13.3) | 10 (35.7) | 1 (3.6) | 0.005* | 3 (13.0) | 1 (3.4) | 0.310 | 2 (10.0) | 3 (13.6) | 1.000 |
| **Pseudomonas aeruginosa** | | | | | | | | | | |
| n (%) | 136 (100%) | 19 (14.0%) | 18 (13.2%) | | 17 (12.5%) | 40 (29.4%) | | 18 (13.2%) | 24 (17.6%) | |

*(Continued)*

**Table 1.** (Continued)

| | Total | Chlorhexidine group | | P-value[a] | Octenidine group | | P-value[a] | Routine care group = control | | P-value[a] |
|---|---|---|---|---|---|---|---|---|---|---|
| | | Baseline | Intervention | | Baseline | Intervention | | Baseline | Intervention | |
| ICU day of sample collection[b], median (IQR) | 12 (6–28) | 14 (4–33) | 15 (9–20) | 0.927 | 6 (5–21) | 15 (9–29) | 0.319 | 11 (5–13) | 12 (5–25) | 0.949 |
| **Sample site** | | | | | | | | | | |
| Blood[b], n (%) | 2 (1.5) | 0 (0.0) | 0 (0.0) | n.d. | 0 (0.0) | 1 (2.5) | 1.000 | 1 (5.6) | 0 (0.0) | 0.429 |
| Tracheal aspirate[b], n (%) | 82 (60.3) | 7 (36.8) | 12 (66.7) | 0.070 | 12 (70.6) | 26 (65.0) | 0.682 | 11 (61.1) | 14 (58.3) | 0.856 |
| Urine[b], n (%) | 16 (7.4) | 2 (10.5) | 2 (11.1) | 1.000 | 2 (11.8) | 4 (10.0) | 1.000 | 2 (11.1) | 4 (16.7) | 0.685 |
| Wound[b], n (%) | 15 (11.0) | 4 (21.1) | 2 (11.1) | 0.660 | 1 (5.9) | 4 (10.0) | 1.000 | 3 (16.7) | 1 (4.2) | 0.297 |
| Others[b], n (%) | 21 (15.4) | 6 (31.6) | 2 (11.1) | 0.232 | 22 (11.8) | 5 (12.5) | 1.000 | 1 (5–6) | 5 (20.8) | 0.214 |

P-values were calculated by Chi-quare test or Fisher's exact test for categorical, and by Mann-Whitney U Test for continuous variables.

[a]P-values were reported for comparisons between baseline and intervention periods for each study group.

[b]P-Values were not shown for baseline period comparisons and intervention period comparisons of the three study groups because no significant differences were detected between the study groups in the baseline and the intervention period. baseline, prior to the intervention period, intervention, at the end of the intervention period.

P-values < 0.05 were interpreted as significant (*).

## Multivariable analyses

We used GEE models to investigate the impact of antiseptic bathing with chlorhexidine or octenidine on the chlorhexidine and octenidine susceptibility of bacterial isolates (all species). Characteristics of bacterial isolates (all species) stratified by chlorhexidine or octenidine MIC $\geq$ species-specific $MIC_{50}$ (yes / no) are shown in S4 Table. aIRR for the chlorhexidine and the octenidine models are depicted in Table 3. The proportion of bacterial isolates with MIC $\geq$ species-specific chlorhexidine and octenidine $MIC_{50}$ did not significantly differ between chlorhexidine or octenidine intervention periods compared with the control group (intervention period, Table 3). We detected significant differences between study groups and study periods with our GEE models (Table 3). However, these differences seemed to be independent from antiseptic bathing. Further, we found evidence that ICU day of sample collection was significantly associated with increased proportions of bacterial isolates with MIC $\geq$ species-specific $MIC_{50}$ for chlorhexidine, but not for octenidine. Interestingly, bacterial isolates with chlorhexidine MIC $\geq$ species-specific chlorhexidine $MIC_{50}$ were significantly less likely to originate from blood compared with other sample sites. This observation was not found for octenidine.

## Discussion

In a large sample of ICU-attributable bacterial isolates collected during the cluster-randomized decolonization trial CLIP-ID, susceptibility to chlorhexidine and octenidine was not reduced 12 months after implementation of daily antiseptic bathing with the respective substances. Some bacterial isolates had enhanced chlorhexidine and octenidine MIC. Most of them were observed in the control group and / or in the baseline period when ICUs did not apply any antiseptic substances. Thus, in our setting, daily bathing with chlorhexidine or octenidine did not enhance the development of non-susceptibilities among bacterial isolates of clinical samples to these substances.

Several *in vitro* and clinical studies suggest an association of high chlorhexidine use and the development of non-susceptibilities of bacterial isolates to chlorhexidine [17–19, 24, 25].

**Table 2. Chlorhexidine and octenidine minimal inhibitory concentrations (MIC) of ICU-attributable bacterial isolates from clinical samples reported by study group and period.**

| | Total | Chlorhexidine group | | Octenidine group | | Routine care group = control | |
|---|---|---|---|---|---|---|---|
| | | Baseline | Intervention | Baseline | Intervention | Baseline | Intervention |
| ***Staphylococcus aureus*** | | | | | | | |
| n (%) | 155 (100.0%) | 38 (25.0%) | 22 (14.1%) | 24 (15.4%) | 18 (11.5%) | 23 (14.7%) | 30 (19.2%) |
| **Chlorhexidine** | | | | | | | |
| MIC$_{50}$ in est. [µg/ml] (IQR) | 5.0 (2.0–5.0) | 5.0 (5.0–5.0) | 2.0 (2.0–4.3) | 5.0 (2.0–5.0) | 2.0 (2.0–5.0) | 5.0 (2.0–5.0) | 5.0 (2.8–10.0) |
| MIC$_{90}$ in est. [µg/ml] | 5.0 | 5.0 | 4.3 | 5.0 | 5.0 | 5.0 | 10.0 |
| **Octenidine** | | | | | | | |
| MIC$_{50}$ in est. [µg/ml] (IQR) | 2.0 (2.0–2.0) | 2.0 (2.0–4.0) | 2.0 (0.8–2.0) | 2.0 (2.0–2.0) | 2.0 (2.0–2.0) | 2.0 (2.0–4.0) | 2.0 (2.0–2.0) |
| MIC$_{90}$ in est. [µg/ml] | 2.0 | 4.0 | 2.0 | 2.0 | 2.0 | 4.0 | 2.0 |
| **Coagulase-negative staphylococci** | | | | | | | |
| n (%) | 122 (100.0%) | 31 (25.4%) | 31 (25.4%) | 17 (13.9%) | 11 (9.0%) | 15 (12.3%) | 17 (13.9%) |
| **Chlorhexidine** | | | | | | | |
| MIC$_{50}$ in est. [µg/ml] (IQR) | 5.0 (2.0–5.0) | 5.0 (2.0–5.0) | 2.0 (1.0–2.0) | 5.0 (5.0–5.0) | 2.0 (1.0–3.5) | 5.0 (2.0–5.0) | 5.0 (5.0–5.0) |
| MIC$_{90}$ in est. [µg/ml] | 5.0 | 5.0 | 2.0 | 5.0 | 3.5 | 5.0 | 5.0 |
| **Octenidine** | | | | | | | |
| MIC$_{50}$ in est. [µg/ml] (IQR) | 2.0 (2.0–4.0) | 2.0 (2.0–4.0) | 2.0 (0.8–2.0) | 2.0 (2.0–4.0) | 0.8(0.8–2.0) | 4.0 (2.0–4.0) | 2.0 (2.0–2.0) |
| MIC$_{90}$ in est. [µg/ml] | 4.0 | 4.0 | 2.0 | 4.0 | 2.0 | 4.0 | 2.0 |
| ***Escherichia coli*** | | | | | | | |
| n (%) | 227 (100.0%) | 64 (28.2%) | 37 (16.3%) | 41 (18.1%) | 18 (7.9%) | 39 (17.2%) | 28 (11.9%) |
| **Chlorhexidine** | | | | | | | |
| MIC$_{50}$ in est. [µg/ml] (IQR) | 5.0 (2.0–5.0) | 5.0 (2.0–5.0) | 2.0 (2.0–5.0) | 5.0 (5.0–5.0) | 2.0 (2.0–5.0) | 5.0 (5.0–5.0) | 5.0 (2.0–10.0) |
| MIC$_{90}$ in est. [µg/ml] | 5.0 | 5.0 | 5.0 | 5.0 | 5.0 | 5.0 | 10.0 |
| **Octenidine** | | | | | | | |
| MIC$_{50}$ in est. [µg/ml] (IQR) | 4.0 (2.0–4.0) | 4.0 (4.0–4.0) | 2.0 (2.0–4.0) | 4.0 (4.0–4.0) | 2.0 (2.0–4.0) | 4.0 (4.0–4.0) | 4.0 (2.0–4.0) |
| MIC$_{90}$ in est. [µg/ml] | 4.0 | 4.0 | 4.0 | 4.0 | 4.0 | 4.0 | 4.0 |
| ***Klebsiella spp.*** | | | | | | | |
| n (%) | 150 (100.0%) | 28 (18.7%) | 28 (18.7%) | 23 (15.3%) | 29 (19.3%) | 20 (13.3%) | 22 (14.7%) |
| **Chlorhexidine** | | | | | | | |
| MIC$_{50}$ in est. [µg/ml] (IQR) | 50.0 (20.0–50.0) | 50.0 (20.0–50.0) | 20.0 (20.0–50.0) | 50.0 (50.0–50.0) | 50.0 (20.0–50.0) | 50.0 (42.5–50.0) | 50.0 (20.0–50.0) |
| MIC$_{90}$ in est. [µg/ml] | 50.0 | 50.0 | 50.0 | 50.0 | 50.0 | 50.0 | 50.0 |
| **Octenidine** | | | | | | | |
| MIC$_{50}$ in est. [µg/ml] (IQR) | 4.0 (4.0–8.0) | 4.0 (4.0–12.0) | 4.0 (4.0–4.0) | 8.0 (4.0–10.0) | 4.0 (4.0–4.0) | 4.0 (4.0–13.0) | 4.0 (4.0–4.0) |
| MIC$_{90}$ in est. [µg/ml] | 8.0 | 12.0 | 4.0 | 10.0 | 4.0 | 13.0 | 4.0 |
| ***Pseudomonas aeruginosa*** | | | | | | | |
| n (%) | 136 (100.0%) | 19 (14.0%) | 18 (13.2%) | 17 (12.5%) | 40 (29.4%) | 18 (13.2%) | 24 (17.6%) |
| **Chlorhexidine** | | | | | | | |
| MIC$_{50}$ in est. [µg/ml] (IQR) | 50.0 (50.0–50.0) | 50.0 (35.0–50.0) | 50.0 (50.0–50.0) | 50.0 (50.0–50.0) | 50.0 (50.0–50.0) | 50.0 (50.0–50.0) | 50.0 (50.0–50.0) |
| MIC$_{90}$ in est. [µg/ml] | 50.0 | 50.0 | 50.0 | 50.0 | 50.0 | 50.0 | 50.0 |
| **Octenidine** | | | | | | | |
| MIC$_{50}$ in est. [µg/ml] (IQR) | 12.0 (8.0–24.0) | 8.0 (8.0–20.0) | 8.0 (8.0–16.0) | 16.0 (8.0–16.0) | 12.0 (8.0–16.0) | 24.0 (8.0–32.0) | 16.0 (8.0–24.0) |
| MIC$_{90}$ in est. [µg/ml] | 24.0 | 20.0 | 16.0 | 16.0 | 16.0 | 32.0 | 24.0 |

Median MIC (MIC$_{50}$) with inter quartile ranges (IQR) and MIC$_{90}$ are given in estimated concentrations (est. µg/ml) derived from extraction solutions that were harvested from the ready-to-use products. MIC are reported for *Staphylococcus aureus*, coagulase-negative staphylococci, *Escherichia coli*, *Klebsiella* spp. and *Pseudomonas aeruginosa* in total and by study group (chlorhexidine, octenidine, routine care) and period (baseline, prior to the intervention period, intervention, at the end of the intervention period). Percentages of bacterial isolates per study group and period refer to the total number per species.

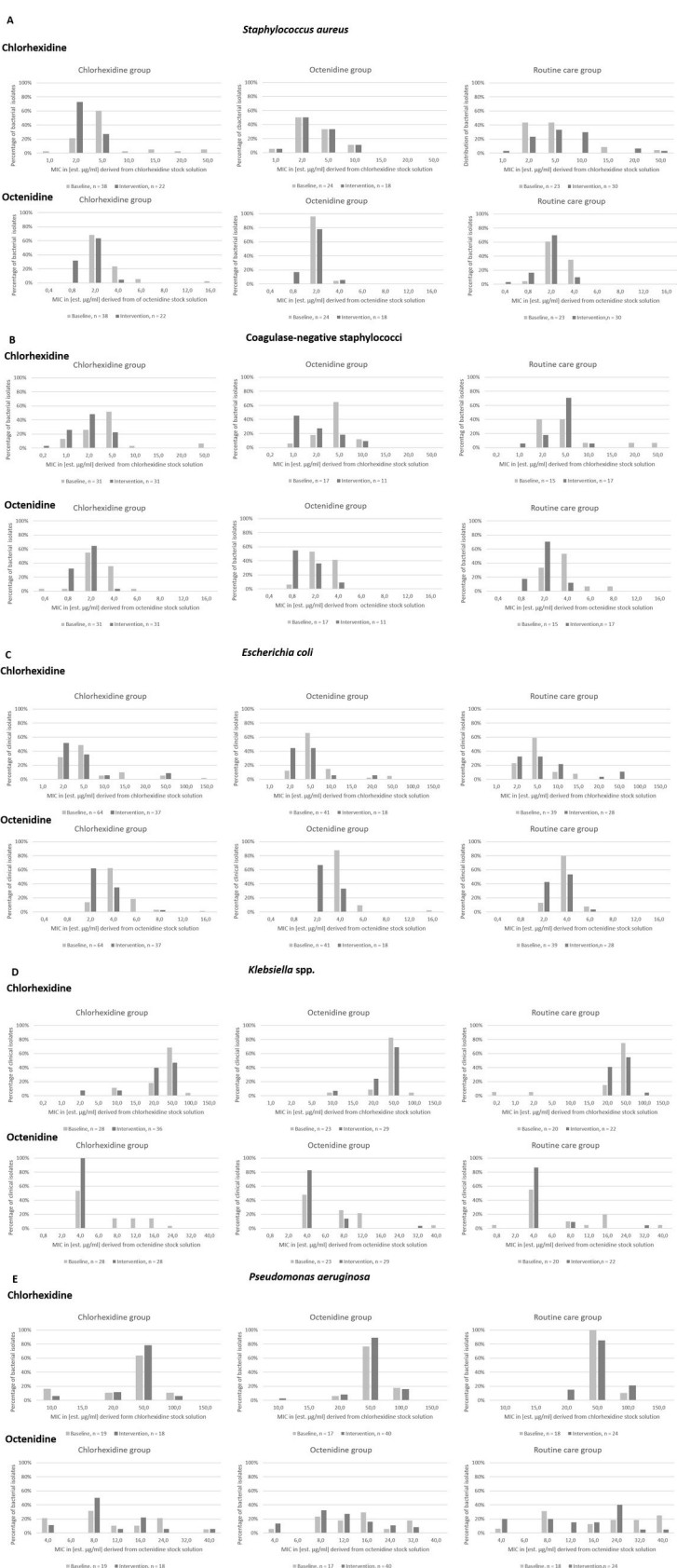

**Fig 2.** Distribution of chlorhexidine and octenidine MIC for all ICU-attributable bacterial isolates from clinical samples with Staphylococcus aureus (A), coagulase-negative staphylococci (B), Escherichia coli (C), Klebsiella spp. (D) and Pseudomonas aeruginosa (E). MIC were reported as estimated minimal inhibitory concentrations (est. MIC) in [μg/ml] derived from the respective extraction solutions. ICU-attributable bacterial isolates from clinical samples were collected prior to (baseline, light grey bars) and at the end of the intervention period (dark grey bars) applying 2%-chlorhexidine impregnated cloths (chlorhexidine group), 0.08% octenidine wash mitts (octenidine group) or water and soap (routine care = control).

However, our findings confirmed data from the REDUCE MRSA trial suggesting that chlorhexidine susceptibility of *S. aureus* isolates was not reduced after implementation of routine chlorhexidine bathing in ICUs [1]. At the same time, data on octenidine susceptibility of bacterial isolates are scarce. Most studies did not report development of non-susceptibility or resistance to octenidine [17, 26–29]. Interestingly, for *P. aeruginosa*, development of tolerances to

**Table 3. Generalized estimating equation models for the outcomes chlorhexidine and octenidine susceptibility of bacterial isolates (reported by MIC $\geq$ species-specific chlorhexidine and octenidine MIC$_{50}$ (yes / no)).**

| *Chlorhexidine* | | | | |
|---|---|---|---|---|
| | aIRR | 95%CI | p-value | p-value (type III) |
| Chlorhexidine group in the baseline period | 0.85 | 0.31–2.35 | 0.771 | < 0.001* |
| Chlorhexidine group in the intervention period | 0.29 | 0.12–0.74 | 0.010* | |
| Octenidine group in the baseline period | 1.81 | 0.69–4.78 | 0.229 | |
| Octenidine group in the intervention period | 0.67 | 0.25–1.79 | 0.428 | |
| Control group in the baseline period | 1.05 | 0.49–2.23 | 0.908 | |
| Control group in the intervention period | 1.00 = reference | - | - | |
| ICU day of sample collection | 1.01 | 1.00–1.03 | 0.013 | 0.013* |
| **Sample site** | | | | 0.015* |
| Blood | 0.36 | 0.19–0.70 | 0.002 | |
| Tracheal aspirate | 0.73 | 0.45–1.17 | 0.187 | |
| Urine | 0.82 | 0.43–1.57 | 0.548 | |
| Wound | 0.79 | 0.41–1.51 | 0.470 | |
| Others | 1.00 = reference | | | |
| *Octenidine* | | | | |
| Chlorhexidine group in the baseline period | 1.27 | 0.56–2.91 | 0.570 | < 0.001* |
| Chlorhexidine group in the intervention period | 0.52 | 0.25–1.07 | 0.085 | |
| Octenidine group in the baseline period | 2.83 | 1.19–6.76 | 0.019* | |
| Octenidine group in the intervention period | 0.68 | 0.32–1.42 | 0.302 | |
| Control group in the baseline period | 1.63 | 0.83–3.23 | 0.160 | |
| Control group in the intervention period | 1.00 = reference | - | - | |
| ICU day of sample collection | 1.00 | 0.99–1.02 | 0.406 | 0.406 |
| **Sample site** | | | | 0.627 |
| Blood | 1.19 | 0.61–2.36 | 0.616 | |
| Tracheal aspirate | 0.95 | 0.55–1.65 | 0.858 | |
| Urine | 0.69 | 0.36–1.64 | 0.276 | |
| Wound | 0.97 | 0.49–1.95 | 0.945 | |
| Others | 1.00 = reference | | | |

Adjusted incidence rate ratios (aIRR) were estimated for all combinations of study groups (chlorhexidine, octenidine, control) and study periods (baseline, intervention) using control group (intervention period) as a reference. ICU day of sample collection, sample sites, cluster effect and interaction between study groups and periods were considered in all models. P-values < 0.05 were interpreted as significant (*). [c] P-values were reported for comparisons between intervention groups (chlorhexidine and octenidine) and control group (routine care), adjusted by study period. aIRR, adjusted incidence rate ratio. 95%CI, 95% confidence interval.

both antiseptics, chlorhexidine and octenidine, has been shown *in vitro* [30]. We observed high chlorhexidine and octenidine MIC for *P. aeruginosa*. In 90% of *P. aeruginosa* isolates, chlorhexidine MIC was ≥ 50 μg/ml that represents the species-specific epidemiological cut-off value (Ecoff) as reported by Kampf et al. [18]. However, no *P. aeruginosa* clinical isolate showed higher MIC than the reference strain.

In our study, the proportions of *S. aureus* and CoNS isolates with elevated chlorhexidine MIC was high (14.8%; 23 of 155 and 8.2%, 10 of 122) when using MIC ≥ 8μg/ml as Ecoff [1, 18]. This observation, however, was independent from the implementation of antiseptic bathing. For *S. aureus* isolates, elevated chlorhexidine MIC (32 μg/ml) and octenidine MIC (3 μg/ml) were reported from hospital trusts in the United Kingdom (UK) after high usage of chlorhexidine and implementation of octenidine usage, respectively [16].

Other studies reported lower proportion of *S. aureus* and CoNS isolates with MIC ≥ 8μg/ml but did not include clinical and / or ICU-attributable isolates [1, 31–33]. In our trial, bacterial isolates were collected from patients who stayed at least 3 days on the ICU. Approximately 70% of ICU patients receive antibiotic therapy [34]. Thus, it is most likely that bacterial isolates from clinical samples in our study were collected from patients previously or currently exposed to antibiotics that might lead to elevated chlorhexidine MIC of their bacterial isolates. However, comparisons to other studies and / or epidemiological cut-off values are only possible to a limited extend. MIC reported here could only be estimated by the concentration of our extraction solutions. Please see limitations for more details.

Our data confirms previous findings that chlorhexidine is highly active against gram-positive and shows lower activity against gram-negative bacteria [35]. Further, estimated MIC required to inactivate bacterial isolates (especially *Klebsiella* spp. and *P. aeruginosa*) was higher for chlorhexidine compared with octenidine. These observations confirm *in vitro* analyses suggesting superiority of octenidine compared with chlorhexidine in the laboratory [36].

The highest estimated chlorhexidine MIC in our study was observed for an *E.coli* isolate (150 μg/ml) collected during the baseline period. The concentration of chlorhexidine in 2% chlorhexidine-impregnated cloths (= 20.000 μg/ml) is more than 130 times higher. In consequence, the concentration applied to the patient's skin during antiseptic bathing is expected to be sufficient (20). For octenidine, the highest estimated MIC in this analysis was 40 μg/ml for *P. aeruginosa* and *Klebsiella spp*. The octenidine concentration in the 0.08% octenidine (= 800 μg/ml) wash mitts is only 20 times higher, but might still be adequate to reduce the bacterial load on the patient's skin. Unfortunately, no studies investigating the biocide concentrations that remain on the patient's skin after bathing with these antiseptic ready-to-use products are available.

Our GEE model found evidence that length of ICU stay (represented by ICU day of sample collection) was associated with higher proportions of bacteria with increased chlorhexidine MIC. Patients with high length of stay on the ICU most likely received more interventions (e.g. invasive procedures, medication, antibiotic therapy) over longer durations of time. As antibiotic resistance is a known indicator of chlorhexidine susceptibility, longer ICU stays (and in consequence presumably more antibiotic treatment) might have an impact on the chlorhexidine susceptibility of bacterial isolates from clinical samples (37). Further, bacterial isolates with high chlorhexidine MIC were significantly less likely to originate from blood cultures compared with other sources. We can only speculate on the reasons for this finding. One possible explanation might be that bacteria isolated from blood might be less affected by ICU interventions including antiseptic bathings compared with other sample sites, e.g. wounds. However, it is unclear, why we did not see these observations with octenidine.

## Strengths and limitations

This is one of the largest multicenter studies investigating the susceptibility of bacterial isolates from clinical samples to the antiseptic substances chlorhexidine and octenidine. Bacterial isolates from 60 different ICUs were analyzed, the majority (70.0%) of them sent isolates before and at the end of the intervention period. Further, we included clinical isolates from patients with length of stay on ICU of at least 3 days. Thereby, clinical isolates were collected from patients receiving all treatments applied in the ICU including the respective bathing procedures (chlorhexidine-impregnated cloths, octenidine wash mitts or water and soap) for at least 3 days. Median ICU day of sample collection was 9 days for all species. In consequence, clinical isolates included in this analyses can be designated as ICU-attributable.

Our study has some limitations. First, the number of bacterial isolates tested per species and study group was low. Second, the observation period of 12 months might be too short to draw any final conclusions. In consequence, results must be interpreted with caution. Further, epidemiological cut-off values for chlorhexidine were not available for all species (e.g. CoNS), and not available at all for octenidine (18). Thus, we used species-specific $MIC_{50}$ as threshold to generate binary variables of phenotypic susceptibilities to chlorhexidine and octenidine for our GEE models. This approach was chosen to apply the same methods for all species and substances (chlorhexidine and octenidine). However, we did not collect all bacterial isolates from clinical samples or a representative sample set from all ICUs participating. Thus, statistical analyses that could be performed to control for potential confounders including cluster-effects were not fully applicable. Fourth, we did not collect any patient-specific data in our analyses (e.g. age, antimicrobial therapy, medication). Potential differences between patient populations of different ICUs could not be considered. Fifth, we did not perform any molecular analyses on the presence of qac genes, efflux pumps, antiseptic resistance genes or any analyses on antimicrobial susceptibility of bacterial isolates. However, this was not subject of our analyses. Several studies are available that investigate the correlation of biocide susceptibility with the absence or presence of efflux pump genes such as qac genes [1, 16, 25, 31, 32]. The correlation of antimicrobial and antiseptic susceptibility is known and has been shown for bacteria such as methicillin-resistant *S. aureus* (MRSA) and Extended spectrum Beta-Lactamase (ESBL) producing *Enterobacterales* [31, 37, 38]. Sixth, MIC comparisons to other studies are only possible to a limited extent as no standardized methods for testing phenotypic susceptibility to the antiseptic substances chlorhexidine and octenidine are available [16, 20]. Even though, MIC and minimal bactericidal concentrations (MBC) are most frequently used in this context, no consensus breakpoints for determining susceptibility to chlorhexidine or octenidine exist [16, 20, 25]. Further, MIC tests were not performed with pure chlorhexidine and octenidine substances but with extraction solutions harvested by wringing out the antiseptic products. Pure stock solutions were not provided by the manufacturers. In consequence, MIC reported here could only be estimated from the concentrations reported for the antiseptic products (2μg / ml for chlorhexidine and 0.08 μg/ml of octenidine) and the dilutions (MIC in percentages) derived from of our extraction solutions. This might also limit comparisons to other studies. However, the main focus of our study was whether susceptibilities of bacterial isolates from clinical samples to chlorhexidine and octenidine might decrease from baseline to intervention periods as well as between antiseptic bathing and control group(s). The method of harvesting antiseptic solutions for MIC tests does not have an impact on these results. In fact, using extraction solutions harvested from the antiseptic products is even more close to reality as it included all additional detergents and ingredients from the ready-to-use product that might reduce the properties of pure chlorhexidine and octenidine substances. Further, this approach might consider fractions of chlorhexidine and octenidine remaining in the cloths / mitts during the

bathing process. This would not be factored in when using pure stock solutions. Thus, we consider our approach being appropriate and even more conservative. However, our estimated MIC might be overestimated and true MICs of bacterial isolates from clinical samples to chlorhexidine and octenidine might be lower. Finally, MIC of bacterial isolates from clinical samples were determined on ICU- not individual patient level. The longitudinal analyses of bacterial isolates from individual patients before, during and at the end of their ICU stay would allow to directly investigate the potential development of any non-susceptibilities of bacterial isolates from clinical samples to chlorhexidine or octenidine during routine antiseptic bathing. Such elaborate sample collections were conducted for analyses of the skin microbiome including antibiotic resistance genes in this project. Results are currently in preparation. Our study has a high likelihood of being underpowered to show smaller, but also clinically relevant increases in the proportions of non- or less susceptible bacterial isolates from clinical samples (all species) to chlorhexidine or octenidine. However, we discussed limitations in detail and add important data on chlorhexidine and octenidine susceptibilities of bacterial isolates from clinical samples to the scarce literature available on this topic.

## Outlook

More analyses on susceptibility of bacterial isolates from clinical samples to antiseptic substances frequently used such as chlorhexidine or octenidine are needed. A standardized protocol would be necessary to compare MIC between different settings and to interpret own findings.

## Conclusions

We found no evidence for reduced chlorhexidine or octenidine susceptibilities of bacterial isolates from clinical samples in ICUs after implementation of daily patient bathing with these antiseptics. However, the observation period of 12 months might be too short and the number of bacteria per species and study group too small to draw any final conclusions. In consequence, results must be interpreted with caution.

## Supporting information

**S1 Data. Data of all bacterial isolates from clinical samples included (n = 790).**
(XLSX)

**S1 Table. Reference strains with minimal inhibitory concentrations (μg/ml) to chlorhexidine and octenidine.**
(DOCX)

**S2 Table. Chlorhexidine and octenidine minimal inhibitory concentrations (MIC) of bacterial isolates from clinical samples in percentage of extraction solution.** Median MIC ($MIC_{50}$) with inter quartile ranges (IQR) and $MIC_{90}$ are given in in [%] of extraction solution and are reported for *Staphylococcus aureus*, coagulase-negative staphylococci, *Escherichia coli*, *Klebsiella* spp. and *Pseudomonas aeruginosa* stratified by study group (chlorhexidine, octenidine, routine care) and period (baseline = prior to the intervention period, intervention = at the end of the intervention period).
(DOCX)

**S3 Table. Characteristics of *S. aureus* isolates from clinical samples with chlorhexidine.**
(DOCX)

**S4 Table. Susceptibility of ICU-attributable bacterial isolates from clinical samples (n = 790) to chlorhexidine and octenidine (all species) stratified by chlorhexidine and octenidine susceptibility.** Chlorhexidine and octenidine susceptibility of bacterial isolates were reported as binary variable (chlorhexidine / octenidine MIC of bacterial isolates $\geq$ species-specific chlorhexidine/ octenidine $MIC_{50}$ (yes / no). [a]P-values were reported for comparisons between chlorhexidine/ octenidine MIC50 $\geq$ species-specific chlorhexidine/ octenidine $MIC_{50}$ = "yes" or "no". [b] percentage of columns. [c] percentage of rows. P-values < 0.05 were interpreted as significant (*). n, number. (%), percent.
(DOCX)

**S1 Checklist. CONSORT 2010 checklist of information to include when reporting a cluster randomised trial.**
(DOCX)

## Acknowledgments

We thank all infection control practitioners, nurses, physicians, healthcare staff and technicians from all ICUs and laboratories participating in and supporting the CLIP-ID trial.

## Author Contributions

**Conceptualization:** Luisa A. Denkel, Petra Gastmeier, Christine Geffers.

**Data curation:** Luisa A. Denkel.

**Formal analysis:** Luisa A. Denkel, Frank Schwab.

**Funding acquisition:** Petra Gastmeier, Christine Geffers.

**Investigation:** Luisa A. Denkel, Jennifer Golembus, Solvy Wolke.

**Methodology:** Luisa A. Denkel, Tobias S. Kramer, Frank Schwab.

**Project administration:** Luisa A. Denkel.

**Resources:** Petra Gastmeier.

**Supervision:** Petra Gastmeier, Christine Geffers.

**Validation:** Luisa A. Denkel, Jennifer Golembus, Solvy Wolke.

**Visualization:** Luisa A. Denkel.

**Writing – original draft:** Luisa A. Denkel.

**Writing – review & editing:** Tobias S. Kramer, Frank Schwab, Jennifer Golembus, Solvy Wolke, Petra Gastmeier, Christine Geffers.

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
