## [Decision Letter · Decision Letter 0]

12 Sep 2022

PONE-D-22-18018Chlorhexidine and octenidine susceptibility of bacterial isolates from clinical samples in a three-armed cluster randomised decolonisation trialPLOS ONE

Dear Dr. Denkel,

Thank you for submitting your manuscript to PLOS ONE. After careful consideration, we feel that it has merit but does not fully meet PLOS ONE’s publication criteria as it currently stands. Therefore, we invite you to submit a revised version of the manuscript that addresses the points raised during the review process.

We look forward to receiving your revised manuscript.

Kind regards,

Surbhi Leekha

Academic Editor

PLOS ONE

Journal Requirements:

   "We thank all infection control practitioners, nurses, physicians, healthcare staff and technicians from all ICUs and laboratories participating in and supporting the CLIP-ID trial. We are grateful to Sage Products 17/ Stryker and Schülke as manufacturers of the (antiseptic) products applied in this trial, for their support. We thank Dr. Brill + Partner GmbH, Hamburg, Germany for their contract work."

 "Funding for the CLIP-ID trial was provided by the German Federal Ministry of Education and Research within the scope of the InfectControl consortium (03ZZ0807A). Sage Products / Stryker and Schülke funded the (antiseptic) products for the intervention and supported the investigation of tolerances to chlorhexidine and octenidine by an independent accredited contract laboratory. The German Ministry of Education and Research (funder) and companies (Sage Products / Stryker, Schülke) funding products and the investigation of tolerances to chlorhexidine and octenidine by an independent laboratory had no role in study design, data collection and analysis, data interpretation, the decision to publish, or the preparation of the manuscript."

4. We note that the original protocol that you have uploaded as a Supporting Information file contains an institutional logo. As this logo is likely copyrighted, we ask that you please remove it from this file and upload an updated version upon resubmission.

Additional Editor Comments:

The manuscript was reviewed by two reviewers with expertise in this area and a statistical reviewer. Please note their comments below with which I agree. Specifically,

1. Please provide the rationale for use of and methods used to determine of the concentration of antiseptic solution generated from wringing out wipes (vs using stock solution)

2. As pointed out by the statistical reviewer, provide power calculations on the effect size for differences in antiseptic susceptibility that could be detected with the target sample size that was already set based on the primary clinical outcome of the trial. The use of more appropriate statistical models is also suggested to statistically compare differences between groups over time.

Minor:

Please clarify what the proportions of different bacterial species shown in Table 2 represent, i.e., what is the denominator?

Please correct the following sentence in the conclusion: “However, the observation period of 12 months might be too and the number of bacteria per species and study group too short to draw any final conclusions.”

Reviewers' comments:

Reviewer's Responses to Questions

**Comments to the Author**

1. Is the manuscript technically sound, and do the data support the conclusions?

Reviewer #1: Yes

Reviewer #2: Partly

Reviewer #3: Yes

2. Has the statistical analysis been performed appropriately and rigorously? 

Reviewer #1: Yes

Reviewer #2: Yes

Reviewer #3: No

3. Have the authors made all data underlying the findings in their manuscript fully available?

Reviewer #1: Yes

Reviewer #2: Yes

Reviewer #3: Yes

4. Is the manuscript presented in an intelligible fashion and written in standard English?

Reviewer #1: Yes

Reviewer #2: Yes

Reviewer #3: Yes

5. Review Comments to the Author

Reviewer #1: Overall, I find this paper interesting with large pool of isolates and interesting set of data which may be used for other researches. I believe this paper should be published, followed some clarifications.

My comments are listed below:

Comment 1: The methods are unclear

1. It is unclear how did you determine the antiseptic stock solutions concentrations of antiseptic substances (chlorhexidine and octenidine), given the fact that they were 'harvested by wringing out'.

2. the production of the Bacterial inoculum is also unclear. It seems that the range of the bacteria CFU is to high 1.5-5.0 108, is it spelling mistake?

Did you meant that the bacterial suspensions were diluted to a final concentration of 10-6 and 10-7? if so, it's need to be better clarify. furthermore, is the bacterial concentration calculated here is before or after the antiseptic and the CSL mixing?

Comment 2: You have a lot of information regarding the isolate origin- e.g blood, urine, tracheal aspirate and liquor- did you looked into differences base on that? If so, show it. It's an interesting data that may contribute largely to current knowledge.

Reviewer #2: The authors performed a study of phenotypic chlorhexidine and octenidine susceptibility of bacterial isolates from clinical samples obtained before and one year after implementation of a three-armed cluster randomized decolonization trial. The results add to the existing literature addressing the impact of antiseptics on resistance patterns. The study design utilized clinical samples and called for the comparison of bacterial isolates obtained prior to initiation of the study with those obtained at the end of the study. This design results in a population-level comparison and the authors acknowledge the limitations of this type of design compared to that of longitudinal patient sampling. The authors also recognize in detail other limitations of their design in the Discussion. Further, the authors inform the readers that a longitudinal patient-level collection of bacterial isolates was done to study the cutaneous microbiome over time, although it’s unclear whether that study will result in comparisons of resistance markers over time in the three arms. It’s also the case that bacteria found in tracheal aspirates, urine, and CSF may have less of an opportunity to be affected by antiseptics used on the skin. That said, the authors provide the results by bacterial species and clinical sample type, which is a strength of their presentation.

Specific Comments/Questions.

1. Abstract, 3rd paragraph of Introduction. Rather than using the terminology “side effects”, something like “unintended consequences” might be a better choice.

2. Methods. Why were stock solutions created by wringing out the impregnated cloths and wash mitts rather than purchasing stock solutions or creating stock solutions from a more concentrated liquid source?

Reviewer #3: A 3-arm cluster randomized clinical trial was conducted which aimed to assess the phenotypic susceptibility of bacterial isolates to chlorhexidine or octenidine for antiseptic bathing. The conclusions are unclear.

Major revision:

More sophisticated statistical models to test for group by time interactions are called for.

Minor revisions:

1- Table 1, title: In statistical terms, stratification refers to a process used during randomization. Consider replacing this term in the title.

2- State and justify the study’s target sample size with a pre-study statistical power calculation. The power calculation should include: sample size, alpha level (indicating one or two-sided), minimal detectable difference and statistical testing method.

3- To assist in the review process, add line numbering to the document.

6. PLOS authors have the option to publish the peer review history of their article (what does this mean?). If published, this will include your full peer review and any attached files.

Reviewer #1: No

Reviewer #2: No

Reviewer #3: No

---

## [Author Response · Author response to Decision Letter 0]

27 Oct 2022

Dear editor, dear reviewers,

Thank you very much for your thorough and helpful comments and advises on our manuscript PONE-D-22-18018 entitled “Chlorhexidine and octenidine susceptibility of bacterial isolates from clinical samples in a three-armed cluster randomised decolonisation trial”. Please find our point by point response in the cover letter entitled PONE_D-22-18018_R1_Rebuttal_final.docx" that we uploaded as additional file. All lines reported in the rebuttal letter refer to the manuscript with track changes.

Comments by editor and reviewers are marked in grey with black font, responses by the authors in blue fonts without marks and changes in the manuscript were marked in yellow with black font. 

We hope that we now fulfill all journal requirements, that we could answer all open questions to your full satisfaction and that we could dissolve all your concerns. Please do not hesitate to contact us, if you need any further information or have any further questions. Thank you very much in advance. 

Kind Regards,

Luisa Denkel

---

## [Decision Letter · Decision Letter 1]

21 Nov 2022

Chlorhexidine and octenidine susceptibility of bacterial isolates from clinical samples in a three-armed cluster randomized decolonization trial

PONE-D-22-18018R1

Dear Dr. Denkel,

We’re pleased to inform you that your manuscript has been judged scientifically suitable for publication and will be formally accepted for publication once it meets all outstanding technical requirements.

Kind regards,

Surbhi Leekha

Academic Editor

PLOS ONE

Additional Editor Comments (optional):

Reviewers' comments:

Reviewer's Responses to Questions

**Comments to the Author**

1. If the authors have adequately addressed your comments raised in a previous round of review and you feel that this manuscript is now acceptable for publication, you may indicate that here to bypass the “Comments to the Author” section, enter your conflict of interest statement in the “Confidential to Editor” section, and submit your "Accept" recommendation.

Reviewer #3: All comments have been addressed

2. Is the manuscript technically sound, and do the data support the conclusions?

Reviewer #3: Yes

3. Has the statistical analysis been performed appropriately and rigorously? 

Reviewer #3: Yes

4. Have the authors made all data underlying the findings in their manuscript fully available?

Reviewer #3: Yes

5. Is the manuscript presented in an intelligible fashion and written in standard English?

Reviewer #3: Yes

6. Review Comments to the Author

Reviewer #3: (No Response)

7. PLOS authors have the option to publish the peer review history of their article (what does this mean?). If published, this will include your full peer review and any attached files.

Reviewer #3: No

---

## [Editor Report · Acceptance letter]

1 Dec 2022

PONE-D-22-18018R1 

Chlorhexidine and octenidine susceptibility of bacterial isolates from clinical samples in a three-armed cluster randomised decolonisation trial 

Dear Dr. Denkel:

I'm pleased to inform you that your manuscript has been deemed suitable for publication in PLOS ONE. Congratulations! Your manuscript is now with our production department. 

Kind regards, 

on behalf of

Dr. Surbhi Leekha 

Academic Editor

PLOS ONE